# Integrating social vulnerability into high-resolution global flood risk mapping

Sean Fox [1] ✉, Felix Agyemang [2], Laurence Hawker [1] & Jeffrey Neal [1,3]

High-resolution global flood risk maps are increasingly used to inform disaster risk planning and response, particularly in lower income countries with limited data or capacity. However, current approaches do not adequately account for spatial variation in social vulnerability, which is a key determinant of variation in outcomes for exposed populations. Here we integrate annual average exceedance probability estimates from a high-resolution fluvial flood model with gridded population and poverty data to create a global vulnerability-adjusted risk index for flooding (VARI Flood) at 90-meter resolution. The index provides estimates of relative risk within or between countries and changes how we understand the geography of risk by identifying 'hotspots' characterised by high population density and high levels of social vulnerability. This approach, which emphasises risks to human well-being, could be used as a complement to traditional population or asset-centred approaches.

Flooding is one of the most common natural hazards associated with climate change globally, with nearly a quarter of the world's population exposed to a 1-in-100-year event[1]. Every year, flooding results in thousands of deaths, the displacement of millions, and hundreds of billions of dollars in damage[2], and climate change is expected to increase both the frequency and intensity of floods in coming decades[3–5]. Exposure to flooding is greatest in low- and middle-income countries (LMICs), and this is expected to increase as a consequence of rapid demographic change[1]. This is of particular concern given that the poorest households and communities have the least coping capacity when confronted with a natural hazard event and suffer the greatest well-being losses[2,6–9]. Failure to mitigate hazard risks for the most vulnerable contributes to the perpetuation of poverty[10] and can exacerbate social inequalities within and between countries[11–13].

Within this context, flood risk mapping is an increasingly essential tool for policymakers seeking to mitigate future risk and respond effectively when hazard events occur[14,15]. In many high-income countries, detailed flood risk maps with multiple indicators of vulnerability have long been used to inform both public policy and private insurance and mitigation[16–19]. By contrast, high-resolution flood risk mapping has been particularly challenging in lower income countries due to basic data deficits[14]. However, recent advances in global flood hazard modelling[20] and population modelling[21,22] have made it possible to produce high-

resolution global flood risk maps for all countries, including those with limited data and modelling capacity. These maps are increasingly used to support disaster risk management, prediction and response by national governments and international organisations[15].

Yet these global flood risk maps do not adequately account for social vulnerability[14], defined broadly as susceptibility to harm from exposure to a hazard event stemming from social factors (e.g. poverty or social status)[23]. Instead, data from hydrological models are combined with gridded population or income and asset (e.g. buildings) data to estimate the amount of people, income or assets likely to be exposed to future flood events[24]. The outputs are more accurately characterised as population or wealth exposure maps than risk maps, as they do not incorporate information about the coping capacity of affected populations. As a result, the places that appear to have the highest risk are either the most densely populated areas or those with a high concentration of income or assets (or both). In short, global flood risk mapping approaches generally ignore social vulnerability and emphasise economic risk, rather than human well-being.

Some recent global studies have attempted to incorporate measures of social vulnerability, such as indicators of poverty[1] or socio-economic resilience[2], but they do so at national scale or for large subnational units (e.g. states, provinces or districts). For example, a recent study estimating the number of low income people exposed to

[1]School of Geographical Sciences & Cabot Institute, University of Bristol, Bristol, UK. [2]Department of Planning, Property & Environmental Management, University of Manchester, Manchester, UK. [3]Fathom, Bristol, UK. ✉e-mail: sean.fox@bristol.ac.uk

flood hazards globally assumes uniform income across large subnational units[1]. Another, which provides a nuanced global analysis of how social vulnerability and resilience intersect with natural hazards to shape risk, only offers national-level estimates[2]. Yet decision makers within local and national governments and international organisations require much higher resolution risk maps to plan for, predict and respond to local flood hazards. The current lack of high-resolution global flood risk maps that account for social vulnerability is therefore an important limitation in current approaches to flood risk analyses that rely on global models.

Here we propose a means of integrating social vulnerability into global fluvial flood risk mapping at 90 m resolution by combining a measure of 'expected population exposure' (EPE) with gridded data reflecting relative deprivation (i.e. poverty), which we use as a proxy measure for social vulnerability in the absence of more detailed and globally available social data. EPE was calculated using annual exceedance probability (AEP) data from Version 2 of the University of Bristol/Fathom Global Flood Model and gridded population data from WorldPop (constrained). We use two alternative sources of data as proxies for social vulnerability: gridded gross domestic product (GDP) per capita transformed into a measure of relative deprivation at cell level, and an index of multidimensional relative deprivation (see 'Methods'). The resulting vulnerability-adjusted risk index (VARI Flood) reflects both the estimated extent of exposure to fluvial flood hazards and the relative vulnerability of potentially affected populations. It can be used to evaluate relative risk at either the global scale (e.g. between countries) or the local scale (e.g. between districts or even neighbourhoods within a district) and can alter how we understand the geography of risk by highlighting areas that are both densely populated and highly vulnerable. Many places that would be considered highly vulnerable due to high population density or high income are found to be relatively less risky when relative vulnerability is considered. The method is particularly useful in national contexts where decision makers faced with scare resources need to prioritise areas for investment in adaptation measures or response in the aftermath of a hazard event. VARI Flood estimates based on country-standardised measures of relative vulnerability help to reveal 'hotspots' of extreme risk for local policy targeting purposes. This approach may therefore serve as a useful complement to traditional population or asset-based measures of risk where decision makers are concerned with mitigating potential well-being losses.

## Results

Our sample is comprised of the 175 countries for which gridded GDP and poverty data are available, collectively containing 98.6% of the global population. In Fig. 1, we show results using both the income-based proxy for social vulnerability (Fig. 1b) and the multidimensional relative deprivation proxy (Fig. 1c). Thereafter we only report results using the income-based proxy given its much greater geographic coverage: globally, the poverty data contain only 11% of the cells contained in the GDP data. In Supplementary Information Fig. S2, we provide evidence that GDP per capita is a reasonable substitute for multidimensional relative deprivation and preferable due to much greater geographic coverage. Full technical details for the main results presented here are provided in 'Methods'.

### Global and regional flood risk

In our sample, approximately 2 billion people are estimated to be exposed to a fluvial flood hazard event using a 10 cm depth threshold. However, to identify the most vulnerable areas, we calculate and compare the populations in the top two quintiles of risk using an exposure-only approach and our VARI Flood method. The income-based relative deprivation measure used for the analyses presented in Figs. 1b and 2 are globally standardised—i.e. reflect quintiles of the global income distribution—to facilitate cross-national comparison.

Using this approach, we estimate that 1.23 billion people live in areas in the top two quintiles of EPE, whereas 866 million people live in areas classified in the top two quintiles of our VARI using the income-based relative deprivation measure—our proxy for social vulnerability. Using the alternative multidimensional relative deprivation index yields lower numbers (867 million and 373 million, respectively) due to limitations in geographic coverage. The differences in these estimates are clear in Fig. 1a and b, which report population counts and shares at admin 2 level for all countries in our sample. The admin 2 unit refers to the second geographic tier of political subdivision within countries, just below the admin 1 units of provinces or states. Admin 2 units are generally counties (in the USA), districts, or other local authority areas and are taken from Fieldmaps (Fieldmaps.io). Visual comparison shows that both the magnitude and the geography of risk changes when social vulnerability is factored into the risk assessment. At the global level, many populous areas are classified as high risk when EPE is the sole criterion but register as low risk when vulnerability is considered. Notable examples include large areas of Australia, Europe, North America, and Russia. At this global scale, the highest risk admin 2-unit changes in 75 (43%) of the countries in our sample when we consider the share of the population in VARI Flood categories 4 and 5 versus EPE categories 4 and 5.

Figure 2 illustrates the differences in the global and regional distribution of risk according to each metric. As Fig. 2a shows for the global sample, an exposure-based approach (i.e. not accounting for social vulnerability) shows a monotonic distribution of risk with over half of the population living in areas classified as high (quintile 4) or very high (quintile 5) risk. By contrast, the VARI Flood estimates show a more normal distribution with most people in areas in the lower three risk quintiles. Figure 2b presents the same comparison broken down my major world region. This shift is a result of the weight placed on socially vulnerable areas (as proxied by low areal GDP). It is important to stress that these estimates use globally standardised population and vulnerability quintiles to reflect relative risk between admin 2 units globally. This information is useful for organisations seeking to compare need across major world regions or countries.

### Country-level flood risk maps

Yet the primary use case for the VARI Flood method is within-country comparison to support national or local decision makers. For this application, exposure and vulnerability quintiles are standardised against country specific population and income distributions. As with the globally standardised approach, incorporating country standardised relative deprivation measures alters our understanding of the geography of risk: the highest risk admin 2-unit changes in 73 (42%) of countries in our sample. A global map of country standardised EPE at admin 2 scale is available for replication in other country contexts (see 'Data availability').

To illustrate, Fig. 3 shows national and subnational results for Nigeria, Pakistan, and Vietnam. These country-level results confirm the global trend: the number and share of people in the highest risk category (5) falls and the geography of risk changes. Comparing the geographic distribution of risk between these approaches, the exposure-only estimates in Nigeria show high and geographically extensive concentrations along the Niger river basin and delta, and in the northern regions—the most densely populated areas of the country. Once social vulnerability is accounted for the broad geographic pattern remains, but there is a reduction of relative risk in the south and increased concentration in the far northeast of the country. In Pakistan the exposure-only and VARI Flood estimates are similar, with a slight increase in the number and share of populations at high risk in southern regions. In Vietnam, both estimates suggest high levels of risk across the country, but the vulnerability-adjusted estimates throw into relief hotspots of particularly high risk (both in absolute and relative terms) in parts of the Mekong River Delta and Central Coast regions.

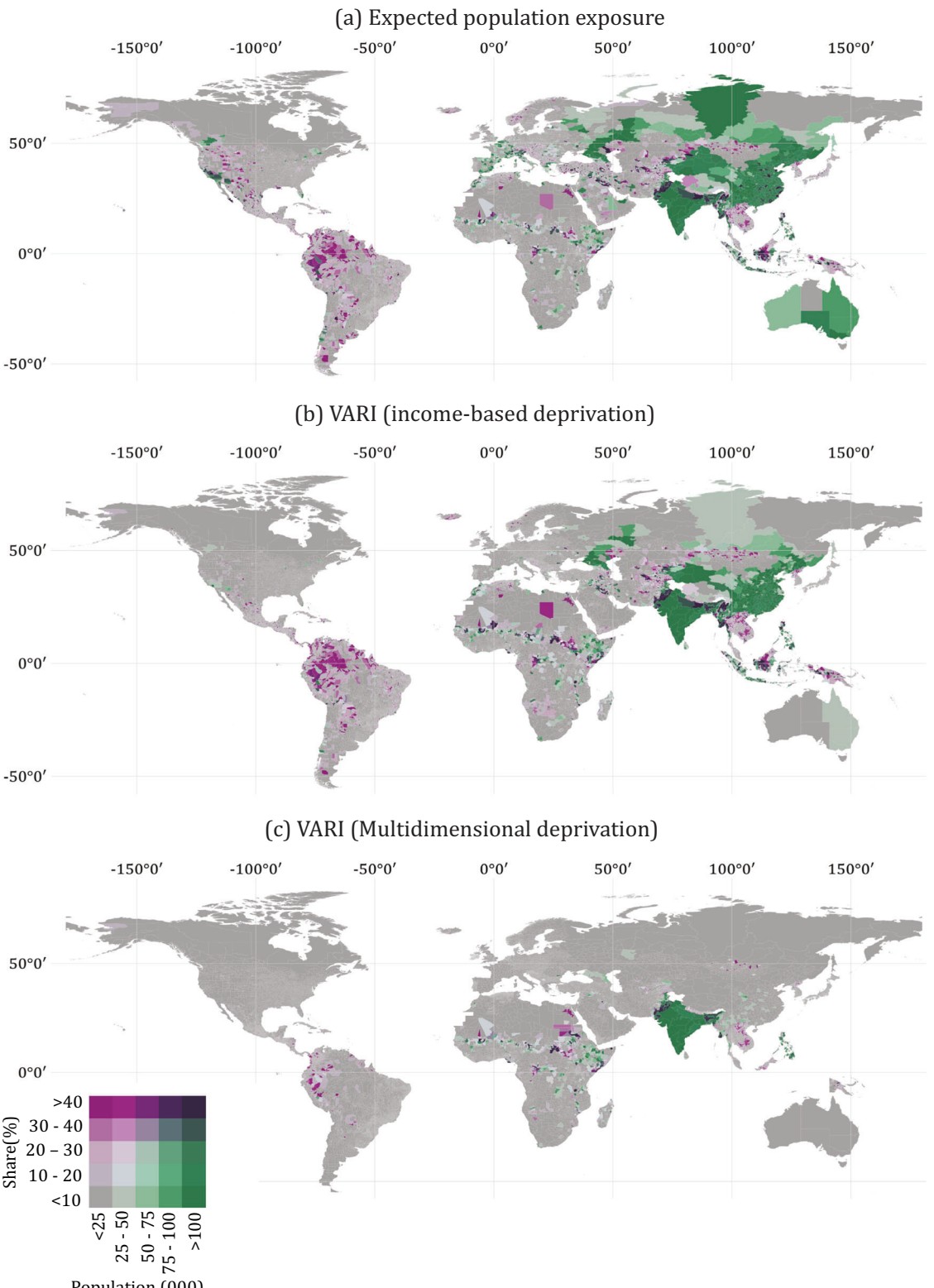

(a) Expected population exposure

(b) VARI (income-based deprivation)

(c) VARI (Multidimensional deprivation)

**Fig. 1 | Global estimates of populations at high risk from flood hazards: a comparison of exposure-only and globally standardised vulnerability-adjusted approaches.** Globally standardised estimates of the size and share of populations living in areas classified in the top two risk quintiles. In **a**, flood risk is measured with reference expected population exposure (AEP × Population) alone; **b** shows vulnerability-adjusted estimates based on quintiles of gridded GDP per capita standardised against the global income distribution; **c** shows vulnerability-adjusted estimates with gridded multidimensional relative deprivation estimates.

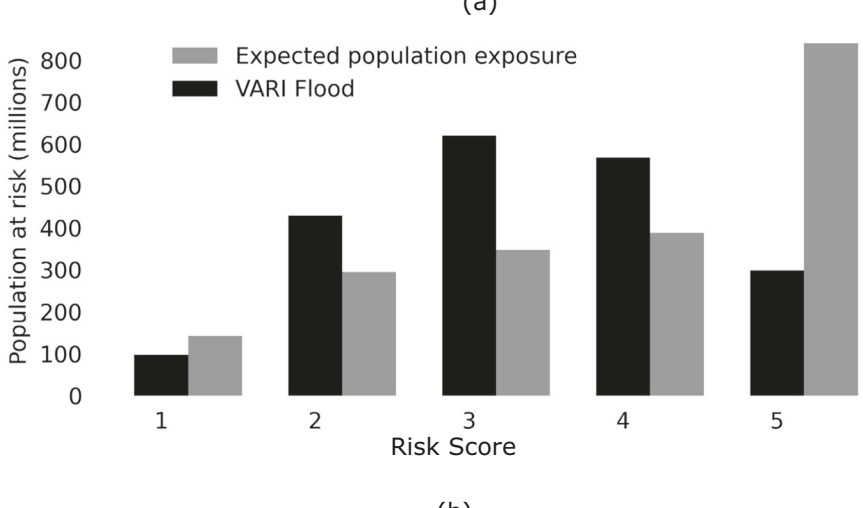

(a)

(b)

| | Expected Population Exposure by quintile (millions) | | | | | Population by VARI Flood by quintile (millions) | | | | |
|---|---|---|---|---|---|---|---|---|---|---|
| | *1* | *2* | *3* | *4* | *5* | *1* | *2* | *3* | *4* | *5* |
| *Africa* | 8 | 20 | 27 | 35 | 80 | 0 | 15 | 40 | 77 | 37 |
| *Americas* | 6 | 17 | 19 | 23 | 77 | 7 | 30 | 48 | 38 | 19 |
| *Asia* | 112 | 236 | 280 | 307 | 642 | 68 | 333 | 496 | 441 | 238 |
| *Europe* | 16 | 21 | 21 | 23 | 39 | 21 | 50 | 34 | 11 | 3 |
| *Oceania* | 0 | 0 | 0 | 0 | 1 | 0 | 1 | 1 | 0 | 1 |
| Total | 142 | 294 | 347 | 388 | 840 | 97 | 429 | 620 | 568 | 298 |

**Fig. 2 | Comparison of global and regional estimates of flood risk estimates with globally standardised measures of relative deprivation. a** shows global estimates of the number of people by flood Risk Score. Relative deprivation quintiles are standardised against the global income distribution to enable cross-regional comparison. Grey bars represent exposure-only estimates based on expected population exposure; black bars represent vulnerability adjusted estimates using GDP per capita as a proxy measure. **b** provides a regional breakdown of estimates using each approach.

Figure 4 drills down further to examine differences at the level of individual human settlements: Amukpe on a tributary of the Benin River in Nigeria, Hyderabad along the Indus River in Pakistan, and Kon Tum on the Dak Bla river in Vietnam. As with both the global and national maps, we observe a noticeable shift in the geography of risk once social vulnerability is factored into our estimates.

These settlement-level risk maps also highlight key limitations of the vulnerability data: it is much lower resolution than the gridded flood model output and population data. While inundation and population can be mapped at high spatial resolution globally (∼90 m), GDP data are only available at 1 km². This is evident in the clear 'tiling' effects, with large squares assigned identical vulnerability values. Although we still capture some variation in risk at the 90 m scale, it may not fully reflect the true variation in risk at this scale given the coarseness of the income data.

## Discussion

Our study presents global gridded fluvial flood risk estimates that explicitly adjust for variations in social vulnerability at 90 m resolution. Incorporating social vulnerability in this way can alter how we understand the geography of risk within countries by emphasising areas with relatively large populations and low coping capacity. This contrasts with traditional approaches that emphasise either population density or income/asset density[25,26].

VARI Flood is designed for decision makers concerned with reducing the impacts of fluvial flood hazards for the most vulnerable by revealing flood risk 'hotspots' where there are likely to be the greatest well-being losses (rather than economic losses). As such, VARI

Flood raises important normative and ethical questions for decision-makers. For example, should flood defences prioritise areas with the greatest number of people, areas with key economic assets and valuable property, or places with the greatest potential well-being losses? How are these imperatives best balanced? By foregrounding potential well-being losses, the VARI Flood approach encourages explicit consideration of these trade-offs.

The results presented here are primarily intended to illustrate an alternative approach to risk estimation with global models rather than reflect definitive estimates of populations at risk within and between countries. These estimates are sensitive to several key assumptions, definitions, uncertainties, and data quality. We assume that poverty is a reasonable proxy for social vulnerability in the absence of more nuanced data (i.e. all else equal, people on lower incomes are generally more vulnerable to stressors than people on higher incomes everywhere). Yet there are many other factors that can affect the vulnerability of individuals, households, and communities. Locally informed vulnerability assessments are preferable where the requisite data are available. Our definition of a 'flood' (i.e. events that exceed 10 cm in depth) may not be suitable in all contexts, particularly where such events are frequent and communities have adapted[27]. Sensitivity analysis presented in Supplementary Information Figs. S3–S6 shows that our estimates are sensitive to the threshold chosen, but also confirms that the VARI Flood approach has a consistent effect with regards to changing our understanding of the distribution of risk.

There are also known uncertainties in the hazard models, which can be large at local scales[28] and likely affect the accuracy of cell-level risk predictions, such as those presented in Fig. 4. However, the

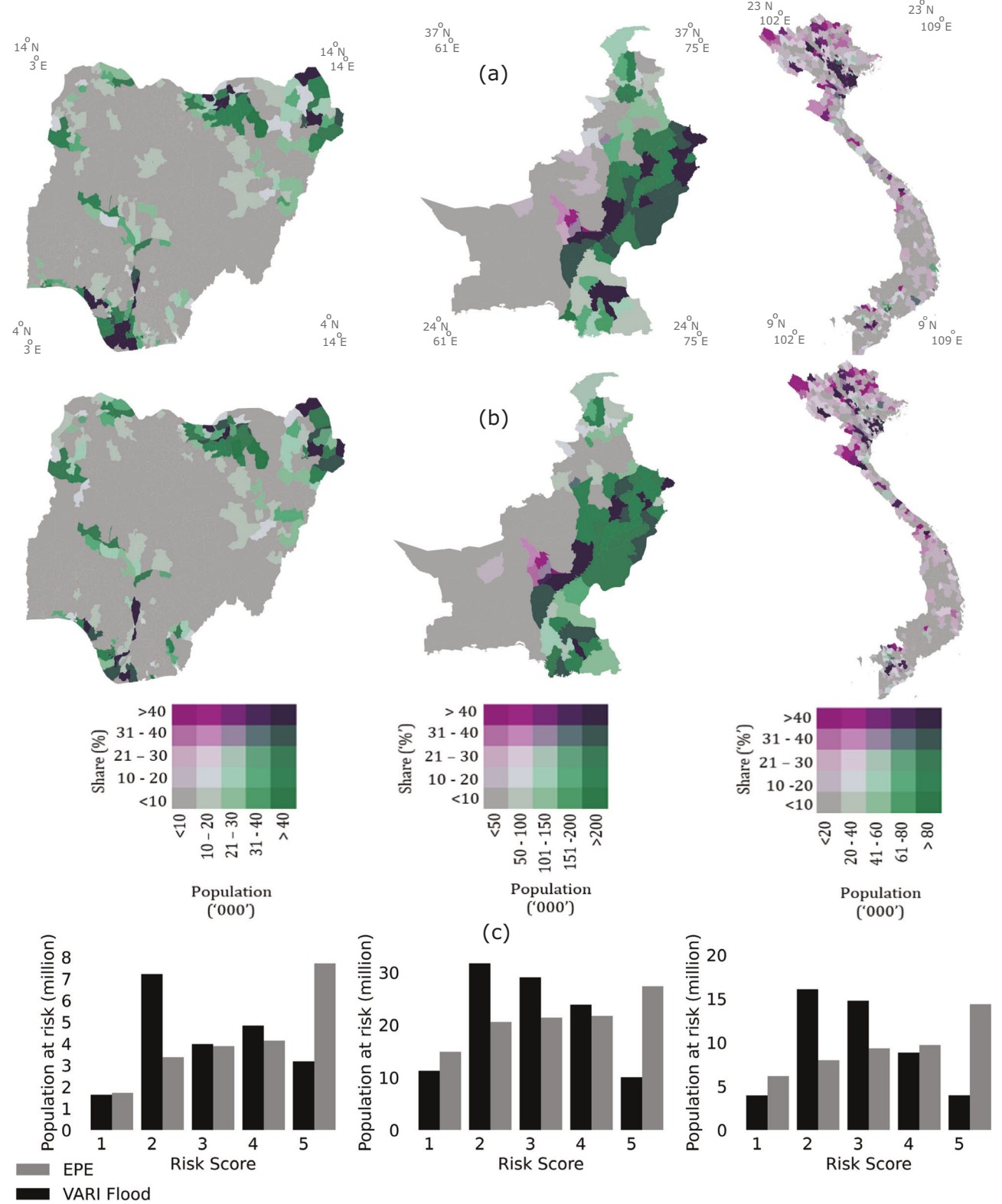

**Fig. 3 | Comparison of EPE and VARI Flood in Nigeria, Pakistan & Vietnam with country standardised measures of relative deprivation. a** shows population counts and shares in the highest risk categories (4 and 5) in Nigeria, Pakistan and Vietnam using expected population exposure; **b** shows population counts and shares in the highest risk VARI Flood categories with relative deprivation standardised against each country's income distribution; **c** compares the country level population counts by quintile using each method.

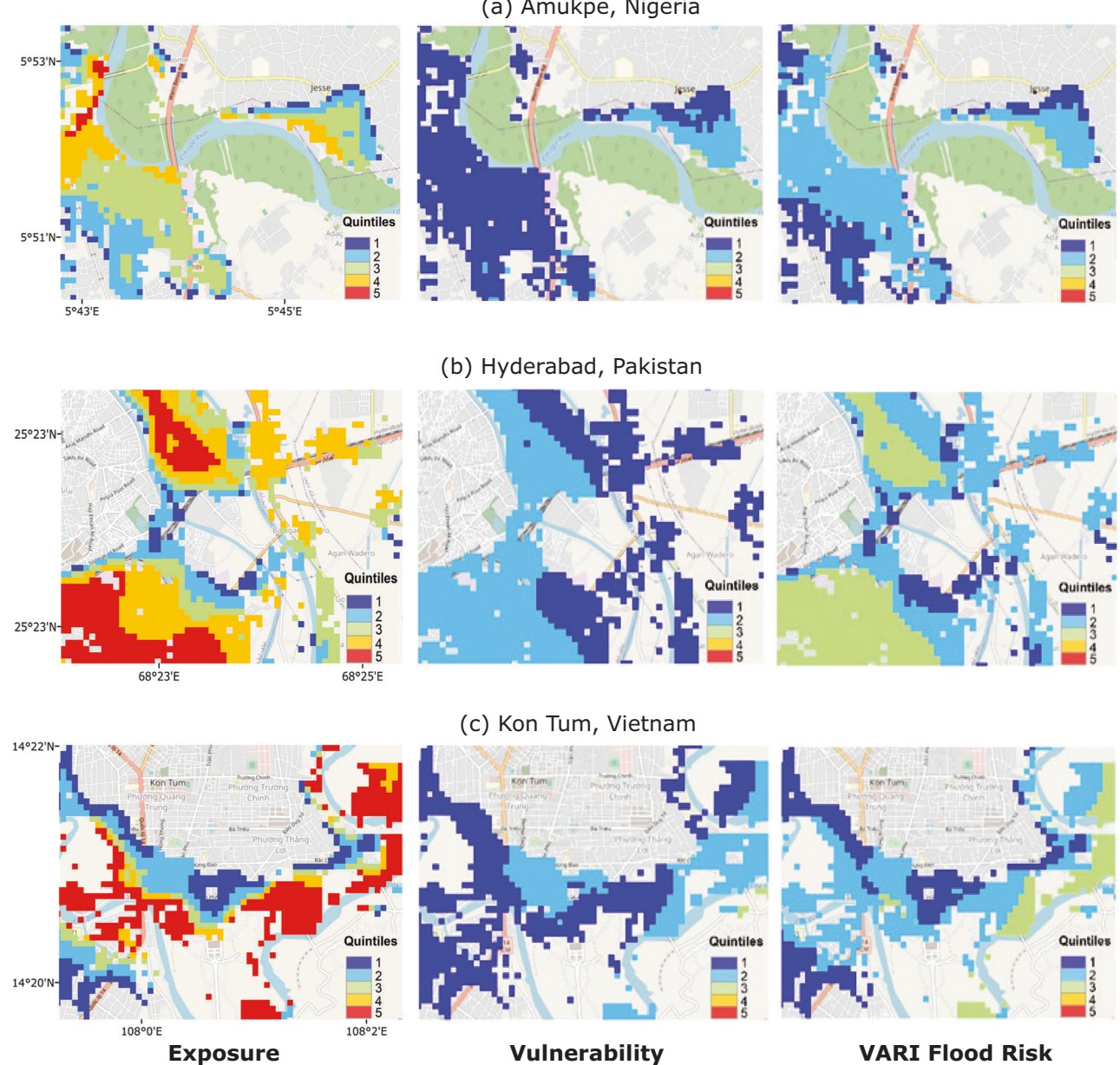

**Fig. 4 | Cell-level comparison of risk estimates within settlements in Nigeria, Pakistan & Vietnam with country standardised measures of relative deprivation.** This figure shows exposure (first column), vulnerability (second column) and VARI Flood (third column) risk scores at the cell level in Amukpe, Nigeria (**a**), Hyderabad, Pakistan (**b**), and Kon Tum, Vietnam (**c**) GDP per capita data are used as a proxy for social vulnerability (see 'Methods').

errors arising from these uncertainties are largely ameliorated when data are aggregated at 1 km² or higher[20], such as the admin 2 unit scale reported in Fig. 3. Nevertheless, investigating how these uncertainties interact with those in the economic data to shape our understanding of the geography of risk is an important direction for future research. Further, we have only estimated fluvial hazards. Incorporating pluvial, small river (<50 km²) and coastal flood exposure, as well as compounding effects between these hazards, would change our risk estimates. However, this would also introduce considerable uncertainty, particularly with regards to pluvial flood models, which are highly sensitive to depth thresholds changes[1] and local topographic features.

Finally, the spatial resolution of our social vulnerability proxies is coarse compared to the available hazard and population data. There is a clear need to substantially improve the resolution of gridded social vulnerability indicators and ensure internal consistency across scales

and geographic contexts. Previous research highlights the often very localised nature of social vulnerability dynamics, which can render a useful indicator at one geographic scale (e.g. district or state) misleading at another (e.g. household or neighbourhood)[29]. An advantage of the VARI approach using relative deprivation is theoretical and empirical consistency, and the ability to calibrate it to any scale of decision-making. But the accuracy of risk estimates will ultimately depend on the quality of the deprivation data and the scale at which it is considered reliable.

The need for higher resolution and higher quality social vulnerability data is particularly acute in rapidly urbanising regions in LMICs. Urban flood risk is growing globally due to demographic and land use changes[30]. According to United Nations projections, nearly all population growth in coming decades will be absorbed by urban areas in Africa and Asia[31]. Low resolution is particularly problematic in large human settlements, such as cities or metropolitan regions, where there

is often considerable local variation in living standards and hence social vulnerability. High-resolution vulnerability-adjusted flood risk maps could help to optimise flood adaptation, mitigation and disaster preparedness investment and intervention in urban areas in these regions.

## Methods

### Measuring social vulnerability with GDP and multidimensional poverty data

Social vulnerability is a multifaceted phenomenon, but can be broadly defined as susceptibility to harm from a combination of exposure to stressors and limited capacity to adapt[23]. There are many factors that affect the vulnerability of individuals, households and communities to natural hazards and these can vary substantially across contexts[23,32]. For example, poverty, age, gender, race and ethnicity, government capacity, and access to emergency services can all influence the extent to which exposure to a hazard event translates into harm, and the extent of harm. Currently, there is no global geospatial dataset that captures the complexity and local nuances of social vulnerability[14], nor variation in the importance of these factors within national and sub-national contexts. However, measures of income and poverty are the most consistent predictors of coping capacity over time and space[29,33]. Simply put, people with lower incomes tend to be more susceptible to harm from stressors than people on higher incomes everywhere in the world. Given the absence of any alternative globally standardised vulnerability data, measures of poverty are the best proxy.

Poverty—or 'deprivation' to use the formal terminology—can be measured in both absolute and relative terms[34]. Absolute deprivation refers to lack of basic needs, such as food, safe drinking water, shelter, etc. This is often measured by using individual or household income as a proxy for the ability to satisfy these basic needs[35]. For example, a common metric for global comparisons is the World Bank's '$1-a-day' measure[1]. However, in recognition of the complex causes of deprivation, multidimensional indices are also common, incorporating measure of health, education, assets, housing conditions, personal security and other 'goods'[35].

These approaches are also applied to measure relative deprivation. Relative deprivation incorporates social context by calibrating measures across populations rather than applying fixed thresholds. For example, one of the key measures of poverty used in the European Union defines a person as relatively deprived if their income is below 60% of national median disposable equivalised income[35]. Multidimensional approaches can include relative health, education, housing, or asset indicators. Relative deprivation indices are particularly useful for policymakers seeking to make decisions about the allocation of scarce resources by identifying people, households, or areas of least and greatest need.

Here we adopt a relative deprivation approach to mapping poverty. The use of relative deprivation rather than absolute deprivation is motivated by the primary use-case for flood risk maps, which is to inform local, national, and international agencies' interventions to help communities adapt and mitigate the consequences of flood hazard events. These actors always operate within a context of scarce resources and must therefore prioritise. At the global level, comparisons can be made across countries by using a globally standardised poverty index that ranks areas based upon where they fall in the global distribution of income, as presented in Figs. 1 and 2 above.

The relative deprivation approach adds greater value at the local scale because globally standardised measures of poverty are not well-suited to local prioritisation. For example, decision makers with scarce resources in Nigeria seeking to mitigate the harm from future flood hazard events will be concerned with identifying the most at-risk areas within their jurisdiction. According to the World Bank, Nigeria is a LMIC. Applying the global LMIC Poverty Line of $3.56 (2017 PPP) per day per capita, over 63% of Nigeria's population lives in poverty[36]—

a figure much too large to facilitate meaningful policy targeting. Conversely, US policymakers at county, state, and federal levels still need measures to identify the most vulnerable places within their jurisdictions, even though less than 1% of the US population lives below the international poverty line. The need for measures that facilitate prioritisation and policy targeting underpins the widespread use by national policy makers of country-specific indicators of relative deprivation[2,35]. Within the context of flood risk assessment, measures of relative deprivation can be useful in highlighting areas of relative social vulnerability and hence elevated risk at various spatial scales.

However, there are currently few global gridded datasets that can be used to measure relative deprivation at high spatial resolution. Chi et al. provide global gridded estimates of relative poverty and wealth at 2.4 km resolution, which is much coarser than existing population data and flood model outputs[37]. To achieve the highest possible resolution and greatest coverage we therefore use two alternative sources. Our preferred data are from Chen et al.[38] who provide gridded GDP at $1 km^2$ resolution globally for 175 countries for the years 1992-2019. These areal estimates of income were converted to per capita estimates by dividing cell GDP by cell population from WorldPop (see below).

For the global results presented in Figs. 1 and 2, we divided cell-level GDP per capita estimates into quintiles using the global income distribution and classified those in the bottom two quintiles as relatively poor and hence relatively vulnerable. For the country-specific results presented in Figs. 3 and 4, we divided cell-level GDP per capita into quintiles using country specific income distributions.

Given the multidimensional nature of poverty and vulnerability, which cannot be fully captured by income data, we also conducted analysis with the Global Gridded Relative Deprivation Index from the Center for International Earth Science Observation at Columbia University[39]. These data are also available globally at $1 km^2$ resolution but offer less comprehensive geographic coverage, containing just 11% of the number of cells available in the GDP data. As demonstrated in Supplementary Information Fig. S2, there is a close cell-level correspondence between our gridded GDP per capita measure and this multidimensional measure. We therefore report results from the GDP-based measure of relative deprivation in the main body of the paper.

### Annual exceedance probability (AEP)

Fluvial flood hazard maps from Version 2 of the University of Bristol/Fathom Global Flood Model (GFM)[20] are used in this study. This version uses the same modelling framework as described in Sampson et al., but with improved elevation data from MERIT DEM[40], a more accurate river network from MERIT-Hydro[41] and an updated method to estimate river conveyance capacity[42]. Model boundary conditions (i.e. how much water is in the river) is estimated using a global regionalised flood frequency analysis as described by[43]. All catchments with an upstream area greater than $50 km^2$ are simulated. Flood depth at 3 arc second spacing (~90 m at the equator) is provided for 10 return periods (5, 10, 20, 50, 75, 100, 200, 250, 500 and 1000). Flood return periods are converted into AEP by first creating a binary flood map where a flood depth >10 cm is considered flooded, then assigning the flooded pixels the AEP value (i.e. inverse of the flood return period in question). For each pixel, the maximum AEP value (i.e. most frequent flood hazard probability) from the stack of flood hazard probabilities is taken, thus creating a single map of flood hazard probabilities.

The choice of a 10 cm threshold for flooding was chosen based on prior literature[25,26]. However, this threshold may not be relevant in contexts where people have adapted to regular events of this magnitude[27]. This low threshold increases uncertainty in our risk estimates as the negative impacts of inundation events becomes increasingly certain as depth increases[44]. In Supplementary Information Figs. S3–S6, we present sensitivity analysis in our three case study countries using alternative depths of 50 and 100 cm. The higher the threshold, the lower the number of people affected. However, the key

conclusion that the distribution of risk changes when vulnerability is accounted for is confirmed. In short, varying our assumptions about the nature of flood hazards does not change our understanding of the nature and geography of risk as much as incorporating a measure of social vulnerability into flood risk assessments.

## Gridded population data

Population information is provided by the building constrained version of WorldPop. This version of WorldPop distributes census information to building locations derived from satellite data using a random forest method and various spatial covariates[45,46]. This data provides population counts at 3-arc second grid spacing for 2020, with estimates adjusted to match United Nations (UN) population estimates.

It is important to note that we chose a 'constrained' population dataset, rather than an 'unconstrained' version (such as WorldPop Unconstrained, LandSan or GPWv4). Unconstrained population datasets differ in that they do not consider building footprints and thus distribute population to all habitable pixels rather than where people reside. This has important implications when considering flood risk as unconstrained population datasets tend to distribute people across uninhabited land (e.g. frequently flooded floodplains)[45,47], whereas in reality people (often) live on the margins of floodplains, out of reach of all but the most extreme floods[27]. A result of this is that unconstrained population datasets overestimate flood exposure[26,48].

## Spatial matching of GDP, AEP and population data

The AEP and population data have similar spatial resolutions, 3 arc-seconds at the equator, which is approximately 90 m using WGS 1984 Pseudo-Mercator projection system. The spatial resolution of the GDP data, however, is 30 arc-seconds at the equator, approximately 1 km. It is important that the spatial resolutions of the three datasets are consistent prior to combining them to construct VARI Flood. We therefore up-sampled the GDP data for each country to spatially match the resolution of the AEP and population datasets using a 'nearest neighbour' technique.

## Constructing a VARI Flood

We combine gridded AEP, population data and GDP per capita data to produce a gridded vulnerability-adjusted risk score for all cells with an AEP > 0. This is done in three stages. First, the AEP for every cell is multiplied by the population of the cell to generate a measure of expected population exposure (EPE) across the full range of return periods. Compared to the traditional approach of using a fixed and arbitrary return period threshold such as a 1-in-100-year flood event[1,3,49], our approach accounts for the fact that hazard events exist on a spectrum of probabilities that cannot be captured by modelling a single probability of flood event (e.g. 1-in-100-year flood event). Small, relatively frequent flooding (<1-in-50-year flood) can result in severe damage and fatalities, such as 2011 Thailand Floods[50]. Recent studies at the sub-national level have found that flood risk inequality is especially prevalent at frequent flood probabilities[30]. On the other end of the spectrum, the likelihood of extreme flood events (>1 in 100) is rising[5,51]. For example, flood risk from notable recent events such as the July 2021 German floods[52,53] and 2017 Hurricane Harvey in the USA[54] would be missed from the typical flood risk assessment using the 1-in-100-year flood. To represent a full range of probabilities, we simulate 10 different flood return periods (5,10,20,50,75,100,200,250,500,1000) of fluvial flooding at 90-metre resolution on rivers with catchment areas above 50 km² and integrate these into a measure of AEP, which is the inverse of return period. This is combined with gridded population data to produce our measure of EPE. The VARI Flood scoring system can be used to evaluate relative risk at multiple scales. For global applications, population and poverty quintiles are drawn from the

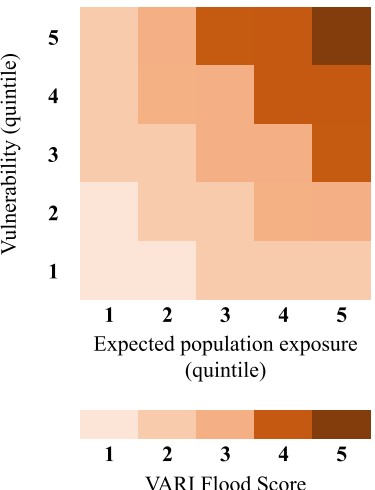

**Fig. 5 | Constructing a vulnerability-adjusted risk index for flooding (VARI Flood).** The index comprises two primary components: (1) expected population exposure (EPE) at cell level, calculated from gridded population data and annual exceedance probability estimates from the flood model; and (2) cell-level measures of poverty as a proxy for social vulnerability. For each cell, we assign a quintile value for EPE (1 = lowest quintile of exposure; 5 = highest quintile of exposure) and vulnerability (1 = least vulnerable quintile; 5 = most vulnerable quintile). The final risk score assigned to a cell is the square root of the product of the EPE and vulnerability quintile values, rounded to the nearest integer. For example, a cell with an EPE quintile value of 4 (i.e. relatively high risk) and a vulnerability value of 5 (i.e. most vulnerable) would receive a VARI Flood score of 4.

global distribution of these variables; at the country level the input quintiles are drawn from the country-specific distributions.

In the second step, the EPE and normalised GDP per capita estimates for all cells are divided into quintiles using the global distribution of these values (for Figs. 1 and 2) and country-specific distributions (for Figs. 3 and 4). We use the inverse of GDP per capita quintiles to reflect social vulnerability, such that the first quintile reflects the richest 20%, and the fifth quintile reflects the poorest 20%. Third, we multiply EPE and vulnerability quintiles, take the square root and round up or down to the nearest integer to get a final risk score (Eq. 1).

$$\text{VARI Flood Score} = \left\lfloor \sqrt{\text{EPE quintile} * \text{Vulnerability quintile}} \right\rceil \quad (1)$$

Values range from 1 to 5, with 1 representing the lowest level of risk (low population and low vulnerability) and 5 representing the highest level of risk (high population and high vulnerability). Figure 5 illustrates how the index is constructed.

This approach differs from past studies by integrating poverty and exposure information to generate a simple risk scoring system. For example, if population exposure at a single return period were the only factor in evaluating the risk of a location (standard practice), the risk index would be highly correlated with population density in places predicted to flood (i.e. more people = more risk). If GDP were used instead, risk would be highly correlated with income (i.e. more income = more risk). The VARI Flood scoring system offers considerably more nuance by factoring in (a) the full spectrum of flood risk (up to 1-in-1000-year probability), (b) population exposure, and (c) relative vulnerability proxied by measures of poverty.

As Fig. 1 makes clear, factoring relative deprivation into risk calculations in this way can substantially affect our understanding of the distribution of flood risk within a population. For example, a traditional approach to risk mapping would assign the highest score (5) to locations in the highest exposure quintile, yet it is possible for such a location to be assigned a VARI Flood score of just 2 if it falls within the

lowest vulnerability quintile. Conversely, it is possible for a location in the lowest exposure quintile to be assigned a VARI Flood score of 2 if it falls within the highest vulnerability quintile. A VARI Flood score of 5 (the highest level of risk) is only assigned to places that fall within the top quintile of both exposure and vulnerability.

At the global level, the index can be calibrated to assess relative risk across countries and regions by taking EPE and GDP per capita quintiles from the global distribution (Figs. 1 and 2). However, the same approach can by applied to assess relative risk within countries by using country-specific population and vulnerability quintiles (Figs. 3 and 4). When constructed in this way, the index can be used to support decision makers working within their national context.

## Reporting summary

Further information on research design is available in the Nature Portfolio Reporting Summary linked to this article.

## Data availability

The raw Fathom global flood model data are restricted for commercial reasons but are available for academic purposes (https://www.fathom.global/contact-us/). The WorldPop constrained high-resolution population counts are available to download (https://hub.worldpop.org/project/categories?id=3). Global gridded GDP are available for download (https://doi.org/10.6084/m9.figshare.17004523.v1). Multi-dimensional relative deprivation data are available for download (https://sedac.ciesin.columbia.edu/data/set/povmap-grdi-v1). Administrative boundary data are from Fieldmaps (https://fieldmaps.io/). Data of subnational flood hazard exposure and risk estimates for 175 countries, which are the output of our analysis, are available for download on Figshare (https://doi.org/10.6084/m9.figshare.25273429.v3).

## Code availability

R was used to prepare annual exceedance probability rasters. Python 3.0 was used for all other analysis. QGIS was used to were used to prepare maps. Replication code for the main analysis is available here: https://doi.org/10.6084/m9.figshare.25285540.v1.

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

## Acknowledgements
This work was funded by the Vietnam National Foundation for Science and Technology Development (NAFOSTED) and UK Natural Environment Research Council (NERC), grant NE/S003061/1 (J.N.). We thank Fathom (https://www.fathom.global) for access to the Fathom Global 2.0 data product.

## Author contributions
S.F. conceived of the idea, developed the formula for calculating risk, and drafted the manuscript. F.A. performed the main analysis and prepared all exhibits. F.A., L.H. and J.N. contributed to conceptualisation, prepared data, and contributed to the final manuscript.

## Competing interests
The authors declare no competing interests.
