## [Peer Review File · Nature Communications]

Integrating social vulnerability into high-resolution global flood risk mappingREVIEWER COMMENTS

Reviewer #1 (Remarks to the Author):

I first would like to congratulate the authors for having conducted an extensive analytical work that combines the more traditional flood risk analysis (i.e., solely using population and/or asset exposure) with social vulnerability considerations, at global level. The topic of and the key message from the research are timely and relevant. Further, the proposed VARI (vulnerability-adjusted risk index) approach is elegant – it is simple, easy to understand, and at the same time have deep meaning and implications.

Despite the innovative work, I still have some reservations for getting this work published due to a number of reasons:

1. Inconsistencies between the main message and the proposed method. To my understanding, the authors' main message is how the geography of flood risk would change at national level when we account for social vulnerability. However, I find the proposed approach of using and quantizing globally gridded GDP can be flawed because of four reasons.

1.1. First, as the authors acknowledge that social vulnerability is more than just income, why not simply use local statistics that often contain much richer set of variables that can be used to picture social vulnerability? Bangladesh has district (adm-2) statistics reports that are publicly available. Indonesia has statistics up to adm-4 level (SMERU Poverty Map). I know that this will not be global – however, with a number of country case studies, the authors will be able to convey the same message as exemplified in the abstract but with a much more robust methodology. The author would still be able to produce something like Figure 4 that shows how the geography of flood risks would change.

1.2. Second, if the goal is to quantify globally how the number of people in the top two quintiles would change, then the approach of classifying GDP at country level to measure social vulnerability is clearly flawed. This simply assumes that the bottom 20% in the USA is just as vulnerable as the bottom 20% in Somalia, which is clearly wrong. Having said that, I don't think that the author should emphasize the global numbers in the paper. For such global numbers, the approach of Rentschler et al 2022 (first reference in the paper), for instance, is better in the sense that the authors used an internationally standardized measure of social vulnerability (i.e., poverty line).

1.3. Third, if the goal is to look at how the geography of flood risk would change at a more local level, then as the author mentioned on line 206-209 and then 235-242, the proposed approach of using globally gridded GDP is not at all justifiable. This is especially true in urban settings where the poor and the vulnerable often live literally side by side with the rich, in which case, not only the 1km grid is insufficient, but also the 90m flood map is too coarse. Having said this, as the author mentioned that urbanization is one key driver of global flood risk and as urban poor becomes a growing concern, I'm wondering on the applicability of the overall approach proposed by the author for capturing social vulnerability in global scale flood risk assessment (as more and more poor and vulnerable people live in urban settings).

1.4. Fourth, I assume that there might be correlations between the globally gridded GDP with population density – as more population tends to result in higher GDP. Except if the population in all grids in a country is the same (which is unrealistic), GDP is a very weak proxy of social vulnerability. Low GDP can simply be a factor of less densely populated areas. GDP per capita would be a somewhat better proxy for social vulnerability, as we then have normalized GDP with the population headcount of the grid.

2. The authors need to justify the use of 10cm as a flood depth threshold for population exposure, which seems to be arbitrary for now. I would advise the authors to perform a quick sensitivity analysis

with varying the flood depth thresholds.

3. I'm not too convinced with the robustness of the current operationalization of VARI. For example, the rounded down squared root implies that $EPE=3$ and $vulnerability=3$ would have the same VARI as $EPE=5$ and $vulnerability=3$ or $EPE=3$ and $vulnerability=5$. In reality, government would definitely prioritize $EPE=5$ and $vulnerability=3$ rather than $EPE=3$ and $vulnerability=3$. Having said that, I would suggest authors to also perform a sensitivity analysis on the operationalization of VARI. One scenario could be not rounding down the squared root value, but rounding up/down to the nearest integer instead (more than 0.5 -> round up, less than 0.5 -> round down). Another way is to treat social vulnerability as a multiplication factor of flood risk, so that $VARI = EPE * (1+Vulnerability)$ where Vulnerability is between 0-1.

Reviewer #2 (Remarks to the Author):

- What are the noteworthy results?

- o When accounting for social vulnerability in terms of GDP, a large portion of the population exposed to high risk appears to have a much lower risk. This analysis identifies the most at-risk individuals with lower coping capacity.

- Will the work be of significance to the field and related fields? How does it compare to the established literature? If the work is not original, please provide relevant references.

- o The global scale analysis is a significant contribution.

- Does the work support the conclusions and claims, or is additional evidence needed?

- o The claim that GDP is a drastically different measure than previous methods needs stronger justification. Also, see next response.

- Are there any flaws in the data analysis, interpretation and conclusions? Do these prohibit publication or require revision?

- o There are a couple of methodological concerns that in my opinion require revision and, at a minimum, some sensitivity analysis. First, this study only considers fluvial flooding. Inclusion of pluvial and coastal flooding, as well as the compound effects of all three types flooding would likely result in substantially different patterns in VARI. Further, the VARI is only based on the most frequent flood event that inundates a particular location. This assumption, like the previous assumption, may significantly underestimate exposure and vulnerability. For example, imagine that two pixels with the same population and GDP characteristics are both inundated 10 cm deep by the 50% annual exceedence probability event; however, the 1% AEP event inundates one of the pixels 20 cm deep and the other 200 cm deep due to differences in hydro-meteorological processes and topographic factors. Based on the methods described in the manuscript, the VARI would be based solely on the 50% event and they would have the same VARI value despite marked differences in the effects of less frequent events. A more comprehensive estimate of VARI would be based on integration of flood intensity across all the events that inundate a particular pixel. I recommend that, at a minimum, the authors perform some sensitivity analysis to examine how much the results change if all inundation events and their frequencies are incorporated in calculating VARI. I also recommend, at a minimum, more discussion of the implications of only considering fluvial flooding.

- Is the methodology sound? Does the work meet the expected standards in your field?

- o The general approach of multiplying an integrated exposure value times a social vulnerability score to assess risk is standard. See concerns above.

- Is there enough detail provided in the methods for the work to be reproduced?

- o Yes.

General comments:

- Further explanation for how GDP is a different measure than income/economic assets in previous assessments: Economic assets and income are used interchangeably when the authors describe how previous work has assessed flood risk (lines 112, 223). GDP is used as the only indicator to measure social vulnerability. GDP seems like a similar economic measure, so I think an explanation of what GDP is, how it is measured, and what makes it different from income or other economic asset measures is needed.
- Describe how the findings and challenges of this paper fit in the broader body of social vulnerability research:
 - The authors provided justification for including only GDP as the sole measure of social vulnerability due to data limitations and previous work that state measures of income and poverty are consistent measures of coping capacity. The difficulty of capturing the variation in significance of vulnerability factors depending on context in lines 279-281 is an important part of the justification of only using GDP. This paper demonstrates the challenges of developing a generalizable set of social vulnerability indicators which fits into the greater body of social vulnerability to natural hazards literature. However, the authors don't express how this challenge fits into the greater body of literature.
 - One of the results was that the risk map loses utility at the local scale because the coarse resolution of the data does not capture local nuances in social vulnerability. This is a commonly seen issue in social vulnerability assessments, and it could be beneficial to mention other work that had the same findings to demonstrate how this work fits into the larger body of social vulnerability work.
- Provide reasoning for assessing relative risk within countries instead of doing a cross-country comparison: It was made clear that the output of the study is an analysis of relative risk within countries, but it was not clear why the relative risk approach was taken. A comparable cross-country analysis seems possible with the available data, and it would seem to provide revealing comparison of risk levels across countries. Comparing the results from both approaches could also provide valuable insight.
- There is no mention of uncertainty and modeling error. At a minimum, the authors should briefly and transparently discuss key uncertainties their implications for their findings. For example, where is uncertainty in relative VARI expected to be greatest? Perhaps in relatively flat areas affected by compound flooding with high spatial heterogeneity in social characteristics?

Line specific comments:

- Line 9 -10: Specify that "these estimates" refers to global flood risk estimates as there are many studies that incorporate social vulnerability in flood risk estimates at national and subnational scales.
- Line 23: Add "a" before "1-in-100 year event."
- Line 29: Consider replacing "against this backdrop" with "Within this context"
- Line 30: Consider splitting this sentence into two – Add period after "occur" and replace "but" with "This"
- Line 34: Same comment as line 9-10; specify global flood mapping approaches
- Line 35: the word "data" is switching between plural and singular
- Line 36: List full phrase the first time an abbreviation is used for GDP
- Line 40: add "economic" before "assets"
- Line 56: suggest not assigning an approximate return period to Harvey, probably much less than 100 yr
- Lines 59-60: Probability expressions are easier for non-experts to interpret than what? Research shows that probability of flooding is also poorly understood by the public
- Line 67: Consider replacing "and these" with "that" to make sentence more clear
- Line 70: Consider explicitly stating that "relative risk within countries" is not a cross country comparison of flood risk. This is mentioned in line 165, but I think having it explicitly stated earlier in the paper will make the rest of the paper clearer to the reader.
- Lines 99 -100: Mention implicit assumption of previous mapping exercises. Are there citations for these studies?
- Line 102: The phrase "perceived" flood risk is used several times in the manuscript, but it is unclear

what this is referring to. Perhaps "understood flood risk" is a different option? (Also in lines 110, 171, 179, 193, 221)

Line 112: Insert "be" in between "to" and "relatively."

Line 124: Insert period after "methods"

Line 133: The phrase "admin 2 level" is used several times, and it is unclear what this is referring to. (Also in lines 138)

Line 134: It is clear how the magnitude of risk changes when considering social vulnerability, but it is less clear what the authors mean by the geography of risk changing. Consider elaborating further.

Line 143: Why does much of India appear to have the same VARI, why does VARI appear to be insensitive to within county differences across a billion people? Is this a concern?

Lines 151 – 152: Looking at Figure 3a, this approach is able to identify people who have high exposure, but low coping capacity from those who have high exposure and high coping capacity. Consider elaborating on this finding and its application to flood management in the discussion.

Figure 2: Unclear what unit "Population" is measured in

Figure 3a: Consider including labels of quintiles with the level of risk

Figure 3b: the authors repeatedly underscore that this approach is for relative comparisons within countries, but this table lumps countries together. Why is it okay to do this?

Line 216: The in-text citation is in a different format than the others and not included in the reference list.

Line 236: List full phrase the first time an abbreviation is used for LMIC

Line 237: List full phrase the first time an abbreviation is used for UN

Line 239: Delete "a" before "particularly"

Line 251: The reference being referred to is missing at the end of sentence.

Line 251: What is the reason that only catchments with an upstream area greater than 50 km² are simulated?

Line 252: Insert space after "simulated."

Line 254: Why were depths less than 10 cm not considered flooded?

Line 267: Delete dash before rather

Line 267: Replace "why" with "where"

Lines 282 – 283: Consider changing word choice for "poor" and "rich"

Line 290: List full phrase the first time an abbreviation is used for PPP; List full phrase the first time an abbreviation is used for HDI

Line 297: Insert "that" between "important" and "the"

Lines 309 – 310: Why was the square root taken?

Reviewer #3 (Remarks to the Author):

The authors tackle an important question, considering flood exposure in the context of people's social vulnerability, i.e. coping capacity. The authors correctly identify a gap in the literature – some existing studies have considered flood exposure along with poverty (but only focused on 1-in-100 year floods), while others have conducted flood exposure assessments without accounting for socio-economic characteristics at all.

While the study has several strengths (e.g. it is well written, concise, and the analysis of 10 different return periods is a useful extension of past studies), unfortunately it does not quite meet the standards of this journal.

The study relies on a somewhat outdated gridded GDP dataset (Kummu et al 2018). There are newer datasets (e.g. Chen et al 2022) which claim higher accuracy, which the authors do not mention. The choice of GDP dataset is crucial for the intended purpose of this study. For instance, the Kummu et al

dataset does reasonably well in capturing areas with dense urbanization and economic activity – yet it often fails to detect poor and more sparsely populated areas, which is problematic for a study that tries to detect poor (flood exposed) areas. The manuscript offers no critical analysis or discussion of the adequacy of the Kummum et al dataset – especially a robustness check would be needed to check how the Kummum dataset performs at the low end of the income distribution and capturing poverty (since this was not an objective of the Kummum et al paper).

The study simply assumes that income is a proxy for vulnerability, but in reality a wide range of factors (apart from income) determine people's vulnerability to disasters. For instance, social groupings, ethnicity, gender, religion, rural vs urban location, access to credit, the quality of infrastructure, and the quality of government support programs (e.g. social insurance schemes) – these are all crucial to reducing vulnerability, yet none of them are reflected in the analysis. While the authors acknowledge that they do not account for these factors, this does not resolve the shortcoming. It would be more accurate to describe the analysis in this study as "income-adjusted" rather than "vulnerability-adjusted". Past studies – e.g. Hallegatte et al 2017, cited in the manuscript – have actually developed a quantitative model to incorporate such diverse factors of vulnerability with global coverage. In short, the study is not robustly capturing vulnerability, which is its key proposition.

The study appears to have a significant technical flaw in defining poverty – unfortunately this alone disqualifies it. Its objective is to identify areas with high vulnerability, proxied by low incomes (i.e. high poverty). It assumes that the bottom 40 percent (bottom 2 quintiles) of the income distribution in each country are vulnerable. This forces the arbitrary result that 40% of people in every country are vulnerable because of their social status. This makes little sense in high-income countries where even those in the bottom 40% have relatively high incomes by international standards (i.e. are not vulnerable according to the author's assumption), as well as having access to high quality social and physical infrastructure. But the assumption also doesn't make sense for the lowest income countries, where much more than 40% of the population live in poverty (e.g. in Malawi, Niger, DRC, Madagascar, South Sudan, Burundi, and Mozambique respectively around 80-90% of the population live in poverty according to the World Development Indicators) – but only 40% of them are classified as vulnerable in this study because poverty is defined based on income quintiles rather than levels. As a consequence, the artificially imposed income quintile definition in this study massively overestimates vulnerability in high-income countries, and massively underestimates vulnerability in poor countries – i.e. the opposite of the study's stated motivation. This is illustrated by Figure 2b, which shows virtually zero flood-vulnerability hotspots in Africa (a continent with major flood and poverty challenges).

Minor points:

- The study claims that it has a spatial resolution of 90 m, but the effective resolution of analysis is 1km (which is the resolution of the Kummum et al 2018 data used in this study). The color blocks in Figure 5, second column, show that the GDP data does not have 90m resolution.
- GDP is not a measure of wealth or assets (which are stock measures), but of a measure of economic activity (which is a flow measure). This is misstated throughout the manuscript.

Chen, J., Gao, M., Cheng, S. et al. Global 1 km × 1 km gridded revised real gross domestic product and electricity consumption during 1992–2019 based on calibrated nighttime light data. *Sci Data* 9, 202 (2022). <https://doi.org/10.1038/s41597-022-01322-5>

Response to reviewers

Reviewer 1

I first would like to congratulate the authors for having conducted an extensive analytical work that combines the more traditional flood risk analysis (i.e., solely using population and/or asset exposure) with social vulnerability considerations, at global level. The topic of and the key message from the research are timely and relevant. Further, **the proposed VARI (vulnerability-adjusted risk index) approach is elegant – it is simple, easy to understand, and at the same time have deep meaning and implications.**

(1) **Inconsistencies between the main message and the proposed method.**

To my understanding, the authors' main message is how the geography of flood risk would change at national level when we account for social vulnerability. However, I find the proposed approach of using and quantizing globally gridded GDP can be flawed because of four reasons.

- 1.1. First, as the authors acknowledge that social vulnerability is more than just income, **why not simply use local statistics that often contain much richer set of variables that can be used to picture social vulnerability?** Bangladesh has district (adm-2) statistics reports that are publicly available. Indonesia has statistics up to adm-4 level (SMERU Poverty Map). I know that this will not be global – however, with a number of country case studies, the authors will be able to convey the same message as exemplified in the abstract but with a much more robust methodology. The author would still be able to produce something like Figure 4 that shows how the geography of flood risks would change.

The reviewer is correct that for some national and local studies, richer data are available. However, the primary aim is to generate consistent global estimates of flood risk at high spatial resolution, particularly in places that do not have any/reliable local statistics. A key application of global flood hazard models is risk assessment in places with significant geospatial data deficits, including many low- and middle-income countries¹⁻³. At present, these assessments do not factor in social vulnerability. Introducing a means of doing so at 90m resolution represents a significant step forward in terms of accounting for social vulnerability in flood risk assessments in data poor contexts. We have made this point much clearer in the Abstract and Introduction with additional references.

- 1.2. Second, **if the goal is to quantify globally how the number of people in the top two quintiles would change, then the approach of classifying GDP at country level to measure social vulnerability is clearly flawed.** This simply assumes that the bottom 20% in the USA is just as vulnerable as the bottom 20% in Somalia, which is clearly wrong. Having said that, I don't think that the author should emphasize the global numbers in the paper. For such global numbers, the approach of Rentschler et al 2022 (first reference in the paper), for

instance, is better in the sense that the authors used an internationally standardized measure of social vulnerability (i.e., poverty line).

Our primary goal was not to quantify the change in the top 2 quintiles globally and recognize our presentation was confusing. We have removed it from the paper. However, given the feedback, we have now computed globally standardized estimates suitable for cross-country comparison and have added this new exhibit. This analysis is now similar to Rentschler et al 2022, but instead of using a single absolute poverty threshold estimated at subnational administrative unit level, we use a new gridded relative poverty product to build up estimates based upon much finer geographic resolution.

1.3. Third, **if the goal is to look at how the geography of flood risk would change at a more local level**, then as the author mentioned on line 206-209 and then 235-242, **the proposed approach of using globally gridded GDP is not at all justifiable**. This is especially true in urban settings where the poor and the vulnerable often live literally side by side with the rich, in which case, not only **the 1km grid is insufficient, but also the 90m flood map is too coarse**. Having said this, as the author mentioned that urbanization is one key driver of global flood risk and as urban poor becomes a growing concern, I'm wondering on the applicability of the overall approach proposed by the author for capturing social vulnerability in global scale flood risk assessment (as more and more poor and vulnerable people live in urban settings).

See reply above to 1.1. Our primary aim is to introduce social vulnerability into global flood risk maps, which are already used to inform local/regional decision-making in data sparse contexts for hazard preparedness and response. We agree that the resolution is sub-optimal, but it is currently the highest possible resolution achievable for global flood risk assessment with publicly available data. Even within large cities, our approach offers improved spatial nuance over administrative units in most contexts. We note in the conclusion that an important goal for next generation global flood risk modelling is increasing the quality and resolution of social vulnerability maps.

1.4. Fourth, **I assume that there might be correlations between the globally gridded GDP with population density** – as more population tends to result in higher GDP. Except if the population in all grids in a country is the same (which is unrealistic), GDP is a very weak proxy of social vulnerability. Low GDP can simply be a factor of less densely populated areas. **GDP per capita would be a somewhat better proxy** for social vulnerability, as we then have normalized GDP with the population headcount of the grid.

This assumption is correct. Our analysis uses GDP per capita. We have made this clearer in the revised manuscript where we present GDP estimates.

2. The authors need to **justify the use of 10cm as a flood depth threshold for population exposure**, which seems to be arbitrary for now. I would advise the authors to **perform a quick sensitivity analysis** with varying the flood depth thresholds.

The 10cm threshold was based on prior literature^{4,5}. In the UK at least this threshold came about because it was similar to the errors in the best available topography data and insurance companies regarded this as the typical front doorstep height for residential buildings. However, the reviewer is correct in that it's otherwise somewhat arbitrary because the depth at which flooding becomes a problem is exceedingly variable and complex (this issue applies to essentially all flood risk studies from global to local). We agree that the most suitable way to reflect this issue is to consider the sensitivity of our results to the assumption, therefore we have now incorporated estimates across three thresholds (10, 50 and 100cm) as a sensitivity analysis in an appendix and a discussion explaining the choice of the threshold for the main results presented in the paper.

3. I'm not too convinced with the robustness of the current operationalization of VARI. For example, the rounded down squared root implies that $EPE=3$ and $vulnerability=3$ would have the same VARI as $EPE=5$ and $vulnerability=3$ or $EPE=3$ and $vulnerability=5$. In reality, government would definitely prioritize $EPE=5$ and $vulnerability=3$ rather than $EPE=3$ and $vulnerability=3$. Having said that, I would suggest authors to also perform a sensitivity analysis on the operationalization of VARI. **One scenario could be not rounding down the squared root value, but rounding up/down to the nearest integer instead** (more than 0.5 -> round up, less than 0.5 -> round down).

We thank the reviewer for this comment. We have revised the rounding procedure to the nearest integer. We have also added discussion around the normative implications of incorporating VARI Flood estimates into decision making processes (i.e. trade-offs between targeting areas of greatest exposure vs greatest vulnerability). We note that the new procedure changes the results, but not drastically. Our headline findings regarding how this approach changes our understanding of the distribution and geography of risk remain unchanged. In sum, our method is sensitive to the issue highlighted, but not sufficiently so to alter our key conclusions, which we believe to be robust.

Reviewer 2

“The global scale analysis is a significant contribution.”

- (1) The claim that GDP is a drastically different measure than previous methods needs stronger justification. Also, see next response.

We do not believe we have made this claim in the paper. Our key contribution is using the data in a novel way—essentially turning it upside down to draw attention to vulnerability as opposed to potentially exposed wealth.

- (2) There are a couple of methodological concerns that in my opinion require revision and, at a minimum, some sensitivity analysis. First, this study only considers fluvial flooding. Inclusion of pluvial and coastal flooding, as well as the compound effects of all three types flooding would likely result in substantially different patterns in VARI.

We thank the reviewer for highlighting the importance of other forms of flooding. We have noted this limitation in the revised manuscript, but we have not

undertaken a full multi-hazard assessment at global scale in the revision for several reasons.

First,, it is worth noting that our definition of a fluvial floodplain, extending to rivers with only 50K km² of upstream area, captures rivers an order of magnitude smaller than those of typical global flood modelling studies (e.g. those undertaken by JRC and University of Tokyo). It's also worth noting that there is no clear definition of what constitutes pluvial and fluvial flooding and making comparisons between estimates from different models is complicated by different minimum catchment sizes. True off floodplain pluvial flooding would be significantly more localised than the smallest rivers included in our fluvial domain, and critically would be mostly too localised to be resolved in the latest global scale DEMs. Therefore, a "pluvial" flood model in this global scale context would cover only flooding from rivers smaller than 50k km². Simulating very localised flooding in a global flood modelling context is difficult given the quality of global rainfall and elevation data and is also very difficult to validate as relevant observations are limited. This results in very high sensitivity to threshold choice⁶. We therefore prefer to take an approach where we include in our domain the hazards we have been able to simulate and evaluate over providing complete coverage of all hazards (see for example discussion of pluvial and fluvial hazards in Hawker et al. (2023)⁷. This way we can also make a clear statement regarding the types of flooding we have included.

We would make a similar statement regarding coastal flooding. At the time of writing there is no global hydrodynamic coastal hazard model. The existing models all take a much simpler bathtub type approach to simulating the inundation that will significantly overestimate hazard and cannot simulate compounding effects. We are aware the upcoming generation of Fathom's global flood model will include coastal flooding, this will obviously increase exposure at the coasts, however a global compound hydrodynamic model for all flood types is not currently available.

Clearly from a wider disaster risk reduction perspective it would be beneficial to include all types of flooding (coastal, groundwater, dam break) in our assessment, and indeed it would be worth considering other hazards such as earthquake, wind and fire. However, in this paper we were keen to investigate the novelty of considering social vulnerability rather than undertake a multi-hazard assessment that would generate a number of additional research questions.

(3) Further, the VARI is only based on the most frequent flood event that inundates a particular location. This assumption, like the previous assumption, may significantly underestimate exposure and vulnerability. For example, imagine that two pixels with the same population and GDP characteristics are both inundated 10 cm deep by the 50% annual exceedence probability event; however, the 1% AEP event inundates one of the pixels 20 cm deep and the other 200 cm deep due to differences in hydro-meteorological processes and topographic factors. Based on the methods described in the manuscript, the VARI would be based solely on the 50% event and they would have the same VARI value despite marked differences in the effects of less frequent events. A more comprehensive estimate of VARI would be based on integration of flood intensity across all the

events that inundate a particular pixel. **I recommend that, at a minimum, the authors perform some sensitivity analysis to examine how much the results change if all inundation events and their frequencies are incorporated in calculating VARI.**

The reviewer is correct that there are more sophisticated methods for characterising the hazard and exposure. An event set modelling approach where 10,000 of years of synthetic events are simulated would allow us to properly characterize the exceedance probability. Previously we have implemented such methods in the US and UK⁸. However, we are unaware of any such data set with global coverage that is in the public domain. We believe the reviewer is suggesting a more limited solution based on integrating flood impacts across the multiple return periods to produce an approximation of annual expected impacts, an example of this in the global flood risk context is provided by Ward et al 2013⁴ and we copy the methods figure form this paper below. This method is useful for estimating damage or loss, however it requires a relationship to be defined between flood magnitude (e.g. depth) and the impact (e.g. financial damage). Defining global depth damage relationships is difficult to do and most relationships are heavily biased towards data from developed nations. We do not believe it is possible at this time to derive a dept-impact relationship with reference to social vulnerability that would improve upon what we have done currently, which essentially assumes all flooding above a particular threshold is a problem.

(4) I also recommend, at a minimum, more discussion of the implications of only considering fluvial flooding.

See response to previous reviewers question and new text in the main body of the paper.

(5) Further explanation for how GDP is a different measure than income/economic assets in previous assessments: Economic assets and income are used interchangeably when the authors describe how previous work has assessed flood risk (lines 112, 223). GDP is used as the only indicator to measure social vulnerability. GDP seems like a similar economic measure, so I think an explanation of what GDP is, how it is measured, and what makes it different from income or other economic asset measures is needed.

We have now incorporated an alternative measure of multidimensional relative deprivation in a new appendix and provide evidence that GDP per capita is a reasonable spatial proxy for this alternative measure (and by extension social vulnerability). We explain in the revised paper that GDP is a measure of income, which is widely used in poverty measurement around the world.

(6) The authors provided justification for including only GDP as the sole measure of social vulnerability due to data limitations and previous work that state measures of income and poverty are consistent measures of coping capacity. The difficulty of capturing the variation in significance of vulnerability factors depending on context in lines 279-281 is an important part of the justification of only using GDP. This paper demonstrates the challenges of developing a generalizable set of social vulnerability indicators which fits into the greater body of social vulnerability to natural hazards literature. However, the authors don't express how this challenge fits into the greater body of literature.

We have now provided this context in the introduction with reference to recent literature.

(7) One of the results was that the risk map loses utility at the local scale because the coarse resolution of the data does not capture local nuances in social vulnerability. This is a commonly seen issue in social vulnerability assessments, and it could be beneficial to mention other work that had the same findings to demonstrate how this work fits into the larger body of social vulnerability work.

We have added a comment and key reference on this point in the Discussion.

(8) Provide reasoning for assessing relative risk within countries instead of doing a cross-country comparison: It was made clear that the output of the study is an analysis of relative risk within countries, but **it was not clear why the relative risk approach was taken**. A comparable cross-country analysis seems possible with the available data, and it would seem to provide revealing comparison of risk levels across countries. Comparing the results from both approaches could also provide valuable insight.

We have incorporated a new cross-country analysis with globally-standardized poverty estimates and significantly expanded our justification for also presenting the country-standardized relative estimates in both the Introduction and Methods sections.

(9) There is no mention of uncertainty and modeling error. At a minimum, the authors should briefly and transparently discuss key uncertainties their implications for their findings. For example, where is uncertainty in relative VARI expected to be greatest? Perhaps in relatively flat areas affected by compound flooding with high spatial heterogeneity in social characteristics?

The reviewer makes a good point, and have added a brief discussion of uncertainties in the Discussion. Regarding the hazard model the largest error sources will be the discharge frequency relationship used in the model^{9,10} followed by the assumptions around the river channel conveyance and the DEM. These errors can cause several meters of vertical error in the simulated water sources and typical fits to remotely sensed imagery and local modelling (neither of which are perfect) are around 0.6 (Critical success index). However, although local errors can be quite high these tend to average over larger domains and its typically recommended to conduct the analysis at the pixel scale and then aggregate the findings to reduce these errors. For example, Sampson et al., 2015¹¹ found that the

global flood model error in flooded areas compared to UK government flood maps was unbiased when aggregated to 1 km resolution.

Regarding compound flooding, this is not one of the first order errors sources for this type of model in our experience. The fluvial model will account for compounding effects between river tributaries in the hydraulic calculations. We do not simulate compound flooding from coastal/fluvial processes beyond considering the average sea levels. This will underestimate the flood hazard close to the coast, however it's worth noting that compounding extreme between coastal and fluvial processes are unusual. For example, for a tropical cyclone the storm surge will typically pass before runoff from the land reaches the coast for most storm tracks. So although a combination of extreme river flows and coastal surge might produce significant compound flooding the probability of this happening is low.

Line specific comments:

Line 9 –10: Specify that “these estimates” refers to global flood risk estimates as there are many studies that incorporate social vulnerability in flood risk estimates at national and subnational scales. *Reworded to clarify.*

Line 23: Add “a” before “1-in-100 year event.” *Done*

Line 29: Consider replacing “against this backdrop” with “Within this context” *Done.*

Line 30: Consider splitting this sentence into two – Add period after “occur” and replace “but” with “This” *Done*

Line 34: Same comment as line 9-10; specify global flood mapping approaches *Done.*

Line 35: the word "data" is switching between plural and singular *Fixed.*

Line 36: List full phrase the first time an abbreviation is used for GDP *Have simplified here with “income”*

Line 40: add “economic” before “assets” *Done*

Line 56: suggest not assigning an approximate return period to Harvey, probably much less than 1000 yr. *Agreed and removed.*

Lines 59–60: Probability expressions are easier for non-experts to interpret than what? Research shows that probability of flooding is also poorly understood by the public *We have removed this sentence.*

Line 67: Consider replacing “and these” with “that” to make sentence more clear . *We have reworded this sentence to make it clearer.*

Line 70: Consider explicitly stating that “relative risk within countries” is not a cross county comparison of flood risk. This is mentioned in line 165, but I think having it explicitly stated earlier in the paper will make the rest of the paper clearer to the reader. *The revised text and exhibits have resolved this issue.*

Lines 99 –100: Mention implicit assumption of previous mapping exercises. Are there citations for these studies? *This has been removed in the course of revision.*

Line 102: The phrase “perceived” flood risk is used several times in the manuscript, but it is unclear what this is referring to. Perhaps “understood flood risk” is a different option? (Also in lines 110, 171, 179, 193, 221). *Thanks for this suggestion. We have changed throughout.*

Line 112: Insert “be” in between “to” and “relatively.” *Done*

Line 124: Insert period after “methods” *Done.*

Line 133: The phrase “admin 2 level” is used several times, and it is unclear what this is referring to. (Also in lines 138) *This is explained in the revised text.*

Line 134: It is clear how the magnitude of risk changes when considering social vulnerability, but it is less clear what the authors mean by the geography of risk

changing. Consider elaborating further. *This is illustrated in the country case studies.*

Line 143: Why does much of India appear to have the same VARI, why does VARI appear to be insensitive to within county differences across a billion people? Is this a concern? *This figure was confusing to interpret as presented and has been removed and replaced with a globally standardised analysis.*

Lines 151 – 152: Looking at Figure 3a, this approach is able to identify people who have high exposure, but low coping capacity from those who have high exposure and high coping capacity. Consider elaborating on this finding and its application to flood management in the discussion. *This comment is unclear, but we hope it has been addressed in our extensive revisions.*

Figure 3b: the authors repeatedly underscore that this approach is for relative comparisons within countries, but this table lumps countries together. Why is it okay to do this? *We've replaced this map with new globally standardised estimates.*

Line 216: The in-text citation is in a different format than the others and not included in the reference list. *Fixed.*

Line 236: List full phrase the first time an abbreviation is used for LMIC *Done.*

Line 237: List full phrase the first time an abbreviation is used for UN *Done.*

Line 239: Delete “a” before “particularly” *Done.*

Line 251: What is the reason that only catchments with an upstream area greater than 50 km² are simulated? *We need some minimum threshold. This was seen as a smallest catchment that could be reliably simulated given the quality of the DEM. Below this size the floodplains tend to be smaller than the cell resolution and would not be resolved in the model. Note that this threshold is much smaller than any other global flood model.*

Line 252: Insert space after “simulated.” *Done*

Line 254: Why were depths less than 10 cm not considered flooded? *We chose a common lower threshold. Ultimately, there is no universal definition of a ‘flood’. We note, however, that a recent paper in Nature Comms that estimates the number of people in poverty exposed to floods uses 15cm as the lowest threshold⁶.*

Line 267: Replace “why” with “where” *Done.*

Line 297: Insert “that” between “important” and “the” *Done.*

Lines 309 – 310: Why was the square root taken? *This is a simple and transparent part of the dimensionality reduction to transform a matrix comprised of three inputs into an index with just five values for our final risk index.*

Reviewer 3

The authors tackle an important question, considering flood exposure in the context of people’s social vulnerability, i.e. coping capacity. The authors **correctly identify a gap in the literature** – some existing studies have considered flood exposure along with poverty (but only focused on 1-in-100 year floods), while others have conducted flood exposure assessments without accounting for socio-economic characteristics at all.

- (1) The study relies on a somewhat outdated gridded GDP dataset (Kummu et al 2018). There are newer datasets (e.g. Chen et al 2022) which claim higher accuracy, which the authors do not mention. The choice of GDP dataset is crucial for the intended purpose of this study. For instance, the Kummu et al dataset does reasonably well in capturing areas with dense urbanization and

economic activity – yet it often fails to detect poor and more sparsely populated areas, which is problematic for a study that tries to detect poor (flood exposed) areas. The manuscript offers no critical analysis or discussion of the adequacy of the Kummu et al dataset – especially a robustness check would be needed to check how the Kummu dataset performs at the low end of the income distribution and capturing poverty (since this was not an objective of the Kummu et al paper).

We thank the reviewer for this suggestion. We have incorporated the newer GDP data, which offers much better spatial coverage. This affects our overall population exposure estimates (e.g. total population exposed to potential event rises from 1.45 billion to 2 billion at 10cm).

- (2) The study simply assumes that income is a proxy for vulnerability, but in reality a wide range of factors (apart from income) determine people’s vulnerability to disasters. For instance, social groupings, ethnicity, gender, religion, rural vs urban location, access to credit, the quality of infrastructure, and the quality of government support programs (e.g. social insurance schemes) – these are all crucial to reducing vulnerability, yet none of them are reflected in the analysis. While the authors acknowledge that they do not account for these factors, this does not resolve the shortcoming.

We agree. However, at present there is no other dataset that could be used to measure the multidimensional nature of social vulnerability with the geographic coverage and spatial resolution of the GDP per capita data. We also believe that our assumption that poorer people are, on balance, relatively more vulnerable than rich people is both theoretically and empirically justified. According to Hallegatte et al (2017, pp 2-3) “poor people are exposed to hazards more often, lose more as a share of their wealth when hit, and receive less support from family and friends, financial systems, and governments.” The authors also show strong correlations between GDP per capita and (a) socioeconomic resilience, and (b) risk to well-being (pp. 9).

We have also conducted further analysis with a new gridded measure of multidimensional relative deprivation¹². Estimates using these data are reported in a new appendix because they have much less comprehensive geographic coverage in many flood-prone areas. This results in generally lower estimates across most risk quintiles. However, the broad findings related to changes in our understanding of the distribution of risk hold.

- (3) It would be more accurate to describe the analysis in this study as “income-adjusted” rather than “vulnerability-adjusted”.

We prefer “vulnerability adjusted” to “income adjusted” to draw attention to the way we are using GDP per capital data in a new way. Previous studies have used GDP per capita as a proxy for income or economic assets at risk (i.e. higher = more risk) while we are inverting that assumption (higher = lower risk).

- (4) Past studies – e.g. Hallegatte et al 2017, cited in the manuscript – have actually developed a quantitative model to incorporate such diverse factors of

vulnerability with global coverage. In short, the study is not robustly capturing vulnerability, which is its key proposition.

The Hallegatte et al 2017 report does not provide results disaggregated sub-nationally. Where such analyses are available, they are at the admin 1 or 2 level. Our approach allows for much higher-resolution risk mapping.

- (5) The study appears to have a significant technical flaw in defining poverty – unfortunately this alone disqualifies it. It’s objective is to identify areas with high vulnerability, proxied by low incomes (i.e. high poverty). It assumes that the bottom 40 percent (bottom 2 quintiles) of the income distribution in each country are vulnerable. This forces the arbitrary result that 40% of people in every country are vulnerable because of their social status. This makes little sense in high-income countries where even those in the bottom 40% have relatively high incomes by international standards (i.e. are not vulnerable according to the author’s assumption), as well as having access to high quality social and physical infrastructure. But the assumption also doesn’t make sense for the lowest income countries, where much more than 40% of the population live in poverty (e.g. in Malawi, Niger, DRC, Madagascar, South Sudan, Burundi, and Mozambique respectively around 80-90% of the population live in poverty according to the World Development Indicators) – but only 40% of them are classified as vulnerable in this study because poverty is defined based on income quintiles rather than levels.

We respectfully disagree that there is a technical flaw in our approach to defining poverty, but found this challenge useful in bringing our approach more tightly into focus. We have revised the manuscript to clarify that we are using a ‘relative deprivation’ approach to defining and measuring poverty. We refer the reviewer to the UN Guide to Poverty Measurement, which outlines various approaches, including the relative deprivation approach¹³.

This is a standard approach globally. For example, in Europe people with less than 60% of median income (country-standardized) are classified as ‘relatively poor’¹³. United Nations SDG indicators 1.2.1 and 1.2.2, which relate to Goal 1: “End poverty in all its forms everywhere,” specifically refer to such national definitions. Hallegatte et al (2017) break down the impacts of hazards on the poor and non-poor, “defined as the bottom 20 percent and the top 80 percent in terms of consumption in each country” respectively (pp. 6)¹⁴.

The benefit of this ‘relative deprivation’ approach is its direct relevance to local decision makers. For example, policy makers, planners or disaster response teams in Nigeria have little use for national or district level comparisons with India. They care about assessing relative vulnerability within and across districts in Nigeria, which is what our country-standardized VARI scores at cell level provide.

- (6) As a consequence, the artificially imposed income quintile definition in this study massively overestimates vulnerability in high-income countries, and massively underestimates vulnerability in poor countries – i.e. the opposite of the study’s stated motivation. This is illustrated by Figure 2b, which shows

virtually zero flood-vulnerability hotspots in Africa (a continent with major flood and poverty challenges).

Our measures based on country-standardized risk estimates cannot be directly compared cross-nationally. As such, Figure 2b was a confusing presentation and has been removed. The fact that there appeared to be so few hotspots in Africa was a function of data presentation rather than underlying analysis or results (as the Nigeria example showed). In the revised manuscript we have also included new globally standardized estimates of relative poverty to facilitate direct cross-national comparison.

(6) Minor points:

- The study claims that it has a spatial resolution of 90 m, but the effective resolution of analysis is 1km (which is the resolution of the Kummu et al 2018 data used in this study). The color blocks in Figure 5, second column, show that the GDP data does not have 90m resolution.

The effective resolution of the exposure estimates is 90m, while the effective resolution of the social vulnerability estimates is 1km. As a result, our risk estimates do vary spatially at 90m. While this is a very significant improvement on previous approaches, we note in the manuscript that the 1km resolution is not ideal for social vulnerability assessment. Future research that upscales these estimates could be easily incorporated into revised global VARI maps.

- GDP is not a measure of wealth or assets (which are stock measures), but of a measure of economic activity (which is a flow measure). This is misstated throughout the manuscript.

This is corrected in the revised manuscript.

REFERENCES

1. Wardrop, N. A. *et al.* Spatially disaggregated population estimates in the absence of national population and housing census data. *Proc. Natl. Acad. Sci.* **115**, 3529–3537 (2018).
2. Carr-Hill, R. Missing Millions and Measuring Development Progress. *World Dev.* **46**, 30–44 (2013).
3. Kuffer, M., Owusu, M., Oliveira, L., Sliuzas, R. & van Rijn, F. The Missing Millions in Maps: Exploring Causes of Uncertainties in Global Gridded Population Datasets. *ISPRS Int. J. Geo-Inf.* **11**, 403 (2022).
4. Ward, P. J. *et al.* Assessing flood risk at the global scale: model setup, results, and sensitivity. *Environ. Res. Lett.* **8**, 044019 (2013).
5. Smith, A. *et al.* New estimates of flood exposure in developing countries using high-resolution population data. *Nat. Commun.* **10**, 1814 (2019).
6. Rentschler, J., Salhab, M. & Jafino, B. A. Flood exposure and poverty in 188 countries. *Nat. Commun.* **13**, 3527 (2022).
7. Hawker, L. *et al.* Assessing the next generation of Global Flood Models in the Central Highlands of Vietnam. *Nat. Hazards Earth Syst. Sci. Discuss.* 1–42 (2023) doi:10.5194/nhess-2023-93.
8. Bates, P. D. *et al.* A climate-conditioned catastrophe risk model for UK flooding. *Nat. Hazards Earth Syst. Sci.* **23**, 891–908 (2023).
9. Smith, A., Sampson, C. & Bates, P. Regional flood frequency analysis at the global scale. *Water Resour. Res.* **51**, 539–553 (2015).
10. Devitt, L., Neal, J., Wagener, T. & Coxon, G. Uncertainty in the extreme flood magnitude estimates of large-scale flood hazard models. *Environ. Res. Lett.* **16**, 064013 (2021).
11. Sampson, C. C. *et al.* A high-resolution global flood hazard model. *Water Resour. Res.* **51**, 7358–7381 (2015).
12. Center for International Earth Science Information Network - CIESIN - Columbia University. Global Gridded Relative Deprivation Index (GRDI), v1 (2010–2020). (2022) doi:10.7927/3xxe-ap97.
13. United Nations. *Guide on Poverty Measurement*. (United Nations Economic Commission for Europe, 2017).
14. Hallegatte, S., Vogt-Schilb, A., Bangalore, M. & Rozenberg, J. *Unbreakable: Building the Resilience of the Poor in the Face of Natural Disasters*. (World Bank, 2017). doi:10.1596/978-1-4648-1003-9.

REVIEWERS' COMMENTS

Reviewer #2 (Remarks to the Author):

I am grateful to the authors for their concerted effort to address my comments and concerns. They have by and large satisfactorily addressed them all with the exception of two minor revisions that I recommend because the authors continue to downplay the potential effects of pluvial and compound flooding. They state in their response:

"there is no clear definition of what constitutes pluvial and fluvial flooding and making comparisons between estimates from different models is complicated by different minimum catchment sizes. True off floodplain pluvial flooding would be significantly more localised than the smallest rivers included in our fluvial domain, and critically would be mostly too localised to be resolved in the latest global scale DEMs."

"a combination of extreme river flows and coastal surge might produce significant compound flooding the probability of this happening is low."

For the record, I would argue that pluvial and fluvial flooding are sufficiently defined to an extent that they can be operationalized in compound flood hazard assessments. This study is valuable even though the models used cannot resolve pluvial and compound flooding; however, I think it is important to acknowledge: 1) that while compound flooding may have a low probability (so does their 1000 year fluvial flood!) compound flooding risks are also increasing and can have severe nonlinear consequences; and 2) the majority of streams / watersheds in the world have drainage areas less than the 50 km² threshold used in this study (see for example basic fluvial geomorphology texts like Leopold, Wolman and Miller (1964)). An implication of these points is that the cumulative effects of pluvial flooding, fluvial flooding in small watersheds <50km², and compound flooding could be substantial. As one example, many of the streams around Houston TX have small watersheds, but when it rains about 1.5 meters as it did with Harvey, extensive compound flooding happens in these small watersheds.

To summarize, my recommendation is accept after the authors add a few sentences to acknowledge this context for the study.

Congratulations to the authors on a valuable contribution.

Reviewer #4 (Remarks to the Author):

See comments to Editors below.

Reviewer #5 (Remarks to the Author):

I think you addressed all the issues raised in the first revision round.

In particular:

- 1.1) you explained why you cannot use local statistics for social vulnerability, and other abstracts and introduction are now clearer
- 1.2) you re-computed the globally standardized estimates of social vulnerability, and now results are suitable for cross-country comparison
- 1.3) you justified why you cannot use subgrid information for GDP
- 1.4) you clarified that the analysis already uses GDP per capita
- 2) you performed a sensitivity analysis of the minimum water depth threshold

3) you updated your rounding method of VARI and showed that the results did not change so to affect the key conclusions

The paper can be published as it is.

Response to reviewers

Reviewer 2

I am grateful to the authors for their concerted effort to address my comments and concerns. They have by and large satisfactorily addressed them all with the exception of two minor revisions that I recommend because the authors continue to downplay the potential effects of pluvial and compound flooding. They state in their response:

"there is no clear definition of what constitutes pluvial and fluvial flooding and making comparisons between estimates from different models is complicated by different minimum catchment sizes. True off floodplain pluvial flooding would be significantly more localised than the smallest rivers included in our fluvial domain, and critically would be mostly too localised to be resolved in the latest global scale DEMs."

"a combination of extreme river flows and coastal surge might produce significant compound flooding the probability of this happening is low."

For the record, I would argue that pluvial and fluvial flooding are sufficiently defined to an extent that they can be operationalized in compound flood hazard assessments. This study is valuable even though the models used cannot resolve pluvial and compound flooding; however, I think it is important to acknowledge: 1) that while compound flooding may have a low probability (so does their 1000 year fluvial flood!) compound flooding risks are also increasing and can have severe nonlinear consequences; and 2) the majority of streams / watersheds in the world have drainage areas less than the 50 km² threshold used in this study (see for example basic fluvial geomorphology texts like Leopold, Wolman and Miller (1964)). An implication of these points is that the cumulative effects of pluvial flooding, fluvial flooding in small watersheds <50km², and compound flooding could be substantial. As one example, many of the streams around Houston TX have small watersheds, but when it rains about 1.5 meters as it did with Harvey, extensive compound flooding happens in these small watersheds.

We would like to thank the reviewer for their support for the manuscript. We absolutely agree with the key point that flood risk will be substantial for catchments below 50 km² and that compounding will in places add to this risk. We certainly would not want to downplay this in the manuscript, but equally, as the reviewer notes, we don't believe it detracts significantly from the novelty or value of the work not to include these additional hazards. To address the reviewer concern that we should not downplay the importance of these other flood processes we adjust the manuscript in ways.

First, we have changed "flood risk" to the more specific "fluvial flood risk" in the abstract, introduction, results, and methods sections. Second, we have noted the river size threshold we have used such that this is clear to the reader "on rivers with catchment areas above 50 km²" (lines 322 & 327 in Methods). Finally, we have expanded comment in the discussion (lines 215-219) to highlight the additional risk we know is missing from a fluvial only analysis: "Further, we have only estimated

fluvial hazards. Incorporating pluvial, small river (<50km²) and coastal flood exposure, as well as compounding effects between these hazards, would change our risk estimates. However, this would also introduce considerable uncertainty, particularly with regards to pluvial flood models, which are highly sensitive to depth thresholds changes¹ and local topographic features.”

Our statement regarding the challenge of defining pluvial verses fluvial risk is predominantly aimed at the global hazard models available to characterise this, which use all use different thresholds for defining pluvial and fluvial components of the hazard, or simply restrict their analysis to very large rivers. In fact, our hazard model includes smaller river systems than any other published global flood model to our knowledge. From a more local operational perspective, the definitions, as the reviewer suggest, are likely sufficient for many practical tasks. Nevertheless, we believe we can address the reviewer main concerns with the adjustment to the text above.

Reviewer 4

The authors suggest that the main aim is to identify, within countries, areas of the greatest relative vulnerability to flooding, for example to help inform allocation of scarce resources in humanitarian and disaster response. Given this aim, the approach that has been adopted seems reasonable.

However, the article still contains efforts to make global comparisons. The critique made by R3 is then very relevant. I think it's the case that the authors are (now) using a separate relative deprivation measure (based on global income quintiles?) for the purpose of these global comparisons. But I felt this needed to be discussed a lot more prominently – for example the section (starting on page 4 of the manuscript) where relative and absolute deprivation measures are discussed should include some explicit discussion of the two different approaches taken (within country and between country) to measuring relative deprivation.

We have made multiple changes to ensure clarity in the final manuscript. These include:

Lines 100-103 at the beginning of the Results section: “The income-based relative deprivation measure used for the analyses presented in Figure 1b and Figure 2 are globally standardized—i.e. reflect quintiles of the global income distribution—to facilitate cross-national comparison.”

Lines 131-133: “It is important to stress that these estimates use globally standardized population and vulnerability quintiles to reflect relative risk between admin 2 units globally.”

Lines 141-143: “Yet the primary use case for the VARI Flood method is within-country comparison to support national or local decision makers. For this application, exposure and vulnerability quintiles are standardized against country specific population and income distributions.”

We have also changed the figure titles and captions for figures 2, 3 & 4 to better communicate the distinction between the globally standardized and country standardized estimates.

We hope these changes are sufficient to ensure clarity in the final manuscript.